# Overview of Meta-Analyses: The Impact of Dietary Lifestyle on Stroke Risk

**DOI:** 10.3390/ijerph16193582

**Published:** 2019-09-25

**Authors:** Emma Altobelli, Paolo Matteo Angeletti, Leonardo Rapacchietta, Reimondo Petrocelli

**Affiliations:** 1Department of Life, Health and Environmental Sciences, Epidemiology and Biostatistics Unit, University of L’Aquila, 67100 L’Aquila, Italy; paolomatteoangeletti@gmail.com (P.M.A.); leonardo.rapacchietta@gmail.com (L.R.); 2Public Health Unit, ASREM, 86100, Campobasso, Italy; reimondo.petrocelli@asrem.org

**Keywords:** overview, meta-analyses, stroke, nutrition, geographical areas

## Abstract

A stroke is one of the most prevalent cardiovascular diseases worldwide, both in high-income countries and in medium and low-medium income countries. The World Health Organization’s (WHO) report on non-communicable diseases (NCDs) indicates that the highest behavioral risk in NCDs is attributable to incorrect nutrition. The objective of our work is to present an overview of meta-analyses that have investigated the impact of different foods and/or drinks in relationship with the risk of stroke events (ischemic/hemorrhagic). The papers to be included in the overview were found in MEDLINE, EMBASE, Scopus, Clinicaltrials.gov, Web of Science, and Cochrane Library and were selected according to the preferred reporting items for systematic reviews and meta-analyses (PRISMA) flow chart. Quality assessment were made according to the AMSTAR 2 scale. This overview shows that all primary studies came from countries with high income levels. This evidence shows that many countries are not represented. Therefore, different lifestyles, ethnic groups, potentially harmful or virtuous eating habits are not reported. It is important to underline how the choose of foods may help reduce the risk of cardiovascular diseases and stroke in particular.

## 1. Introduction

A stroke is one of the most prevalent cardiovascular diseases worldwide. It is estimated that in 2010 there were 11,569,538 ischemic stroke events, 63% of which were in medium and low-medium income countries [1]. In the same year, 5,324,997 hemorrhagic strokes occurred, 80% of which were in medium and low-medium income areas [1]. This difference is similar for mortality, which is significantly lower in high-income countries compared to those of middle/medium-low [1]. It is estimated that in Europe the costs of the disease are around €7775 per patient, with a total cost, in billions, of 64,053 euros [2]. In the United States in 2008 the global costs were estimated to be 62.5 billion dollars, the expenditure forecast for 2050 is about 2.2 trillion dollars [3].

The World Health Organization’s (WHO) report on non-communicable diseases (NCDs) indicates that the highest behavioral risk is attributable to incorrect nutrition, particularly in the WHO European region [4]. 

Numerous meta-analysis studies have been conducted to evaluate the relationship between diet and stroke risk. A meta-analysis by Alexander et al. [5] seems to indicate a protective action resulting from the consumption of cheese. This data is in line with Briggs et al. [6]. Dairy products should probably be consumed as part of a balanced diet in which there is adequate intake of all nutrients within an appropriate calorie count [7,8,9,10,11].

Regarding alcohol use and/or abuse [12], red wine contains polyphenols, including resveratrol, a molecule with not only cardio protective pleiotropic effects, but also neuroprotective, anti-microbial and anti-angiogenetic effects [13]. All this has a positive influence on the prevention of ischemic stroke since it acts on one of the main causes, atrial fibrillation [14]. The same may be extended to moderate consumption of beer [15].

Monounsaturated [16] and polyunsaturated [17] fats have been considered as a valid nutritional support: monounsaturated fatty acids (MUFAs) seems to positively modify lipid structure in patients [18].

In addition, it is important to underline consumption of fruit and vegetables [19]. The Centers for Disease Control and Prevention guidelines recommend the daily consumption of 1.5–2.0 cups of fruit and 2.0–3.0 cups of vegetables [20]. 

Consumption of nuts could have a protective role in decreasing the risk of cardiovascular disease [21,22]. The benefit of the intake of nuts seems to be linked to the composition of polyunsaturated fatty acids (PUFAs) that improve the performance of the cardiovascular system as reported by Del Gobbo et al. [23]. 

Finally, the increased risk of ischemic stroke in women who consume high quantities of sugary drinks is due to the insulin peak resulting in the ingestion of large amounts of glucose [24]. Furthermore, it is important to take into consideration how many of these drinks contain added fructose. Fructose enters the glycolytic pathway downregulating it with the intermediate products of its metabolism (glycerol 3-phosphate and acetyl CoA), thus favoring lipogenesis and accumulation of intramuscular and visceral fat [25].

Tea [26] seems to have a role in stroke prevention, as reported by Arab and colleagues as well as folic acid [27,28,29]. The consumption of whole grains does not present significant results [30]. 

The objective of our work is to present an overview of meta-analyses that have investigated the impact of different foods and/or drinks in relationship with the risk of stroke events (ischemic/hemorrhagic). We considered the meta-analyses based on cohort studies and randomized clinical trials.

## 2. Materials and Methods

Meta-analyses regarding the onset of hemorrhagic and/or ischemic strokes in subjects following dietary regimes with a given food or specific nutritional or nutraceutical support have been considered. In addition, studies investigating secondary prevention of strokes were considered, also in relation to a specific food or nutritional or nutraceutical support.

The papers included in the overview were sought in the last 10 years in MEDLINE, EMBASE, Scopus, Clinicaltrials.gov, Web of Science, and Cochrane Library databases up until 31 December 2018. The search strategy was conducted using the following terms: Stroke OR Strokes OR Cerebrovascular Accident OR CVAs (Cerebrovascular Accidents) OR Cerebrovascular Apoplexy, Cerebrovascular OR Vascular Accident, Brain OR Brain Vascular Accident OR Brain Vascular Accidents OR Vascular Accidents OR Brain OR Cerebrovascular Stroke OR Cerebrovascular Strokes OR Stroke, Cerebrovascular OR Strokes, Cerebrovascular OR Apoplexy OR Cerebral Stroke OR Cerebral Strokes OR Stroke, Cerebral OR Strokes, Cerebral OR Stroke, Acute OR Acute Stroke OR Acute Strokes OR Strokes, Acute OR Cerebrovascular Accident Acute OR Acute Cerebrovascular Accident OR Acute Cerebrovascular Accidents OR Cerebrovascular Accidents, Acute AND “Food” (Mesh)) AND “Meta-Analysis” (publication type). The selection of works was conducted using the preferred reporting items for systematic reviews and meta-analyses (PRISMA) method [31] by two-blinded authors (P.M.A. and L.R.). A methodologist (E.A.) resolved any disagreements. To avoid redundant results, we only considered the last published meta-analysis for the same topic.

The quality of the meta-analyses was assessed using the AMSTAR 2 scale by Shea et al. [32] that evaluates the methodological quality of the meta-analyses (Appendix A). In addition, we evaluated the distribution of primary studies included in each meta-analysis according to six different geographical areas (Australia; Canada; China, Singapore and South Korea; Europe; Japan; and USA) and according to four nutritional patterns and/or product types (eating habits, food, beverage, nutrients). 

## 3. Results

The methodology used is described in the preferred reporting items for systematic reviews and meta-analyses (PRISMA) flow chart (Figure 1). 

The literature search highlighted 189 references (Figure 1). After the exclusion of 131 references, the remaining 58 were analyzed by reading the full text, then 23 were excluded: three did not present the outcome of interest and the other 20 were excluded because they were less recent in respect to those included in the review that presented the same outcomes. In total, 35 articles were selected, of which 27 were meta-analyses based on observational studies and eight were randomized controlled trials (RCTs). Table 1 shows the studies by author and by food considered with the respective dose effects found. Appendix A shows the studies by author with the dose response analysis.

Table 2 and Figure 2 show the distribution of primary studies included in each meta-analysis, according to six different geographical areas (Australia; Canada; China, Singapore and South Korea; Europe; Japan; and USA) and according to four nutritional patterns and/or product types (eating habits, food, beverage, nutrients). 

Graphical summary results of meta-analyses are reported in Figure 3, Figure 4, Figure 5 and Figure 6.

### 3.1. Dairy Products 

Four meta-analyses specifically investigated the use of milk and dairy products. In the work of Mullie et al. [8] it is evident that the consumption of 200 mL of milk does not lead to an increased risk of stroke, while Alexander et al. [5] show that risk reduction appears to border statistical significance. Surprisingly, however, the consumption of cheese seems to reduce stroke risk (Table 1, Figure 3). The latter author has also performed a dose-response analysis which suggests that in total the intake of dairy products is protective against stroke; specifically, the daily consumption of cheese with a range from 0.5–1.5 servings; in particular, an intake of calcium from dairy products of 100–300 mg/dL or above 300 mg/dL also helps to protect (Appendix A). On the other hand, a single meta-analysis investigated the correlation between risk of developing stroke and consumption of butter [9] and did not show a statistically significant increase in risk (Table 1, Figure 3). A paper by Wu et al. was concerned specifically with yogurt consumption, but its outcome was not statistically significant, risk reduction (RR) = 1.02 (0.92–1.13) [33]. This evidence was similar also in the dose-response analysis for quantities below 200 g/day, RR = 1.06 (0.98–1.15) and for quantities above 200 g/day, RR = 0.92 (0.85–1.00) [31]. Instead, the more controversial use of calcium along with vitamin D vs. a placebo shows an RR = 1.20 (1.00–1.43) (Table 1, Figure 3) [34].

### 3.2. Alcohol Consumption

Two meta-analyses have been identified that identify alcohol as a risk factor for stroke [7,12]. It is possible to summarize the effect of alcohol on stroke substantially as a biphasic effect: protective, if consumed within the limits of 1–2 alcoholic units but very detrimental in the case of more than 4 alcoholic units (conventionally, a drink containing 8 mg of ethanol is identified as an alcoholic unit). Specifically, the consumption of alcohol seems to be protective in ischemic stroke when comparing mild and moderate consumption vs. non-drinkers, with an RR = 0.87 (0.81–0.92) (Table 1, Figure 5). As for the impact of alcohol on hemorrhagic stroke, heavy drinkers show a markedly higher risk for the onset of an intracerebral hemorrhage when compared to the occasional drinker, RR = 1.74 (1.45–2.09) (Table 1, Figure 6) [12]. Larsson et al. [12] performed a dose-effect analysis to confirm the above data. The consumption of 1–2 alcoholic units a day has a protective effect against ischemic stroke. On the other hand, consumption of 4 alcoholic units is associated with an increased risk of ischemic or hemorrhagic stroke (Appendix A) [12]. 

Zhang’s meta-analysis also shows how a moderate consumption of alcohol has a protective effect compared to heavy consumption (Table 1, Figure 3) [7]. 

### 3.3. Monounsaturated Fatty Acids (MUFAs) and Polyunsaturated Fatty Acids (PUFAs)

A meta-analysis with 10 cohort studies included [16] investigated the consumption of MUFAs; its results show that RR is at the limits of statistical significance (Table 1, Figure 4). 

Meta-analyses of Abdelhamid [35] and Hooper [36] on RCTs showed that omega-3 and omega-6 do not influence stroke risk, respectively: RR = 1.06 (0.96–1.16) and RR = 1.36 (0.45–4.11). Larsson et al. [17] investigated the consumption of PUFAs, also on cohort studies, finding these molecules to be protective of ischemic stroke (Table 1, Figure 5). On the contrary, Abdelhaimid’s meta-analysis [37] on RCTs showed a non-significant PUFA effect on stroke risk: RR = 0.91 (0.58–1.44).

### 3.4. Saturated Fatty Acids

Muto et al. [38] investigated the effect of a diet rich in saturated fatty acids. They showed that with regard to ischemic stroke, the overall RR was 0.89 (0.82–0.96), while it was 0.68 (0.47–0.96) for hemorrhagic stroke. Not significant results were found in Hooper et al. meta-analysis (Table 1, Figure 5 and Figure 6) [39].

### 3.5. Olive Oil

Martin-Gonzales’ meta-analysis has highlighted that olive oil consumption has a protective effect against stroke: RR = 0.74 (0.60–0.92) (Table 1, Figure 3) [40].

### 3.6. Vitamin E

The results of a meta-analysis by Cheng et al. regarding observational studies, highlighted that vitamin E supplements decrease stroke risk: RR = 0.83 (0.73–0.94) (Table 1, Figure 3) [41]. On the other hand, a meta-analysis on RCTs by Bin et al. [42] showed that vitamin E supplements are irrelevant to stroke onset: RR = 1.01 (0.94–1.07). 

### 3.7. Hazelnuts

Chen [21] investigated the consumption of nuts and the incidence of stroke. The consumption of hazelnuts appears to be protective against stroke (Table 1, Figure 3). There are, however, some differences regarding the consumption of different types of hazelnuts (Table 1, Figure 3).

In the dose-effect study, Chen showed how a weekly consumption of up to five portions could reduce mortality [21] (Appendix A).

### 3.8. Black and Green Tea

A meta-analysis by Arab et al. [26] investigated the consumption of green and black tea as a protective factor against the onset of stroke. The results, shown in Table 1, appear to be rather encouraging, favoring a reduction in the risk of stroke (Table 1, Figure 3).

### 3.9. Sugary Drinks

Narain et al. [43] studied the consumption of sugary drinks, determining how a high intake of such drinks, especially in women, seems to favor ischemic stroke (Table 1, Figure 5).

### 3.10. Whole Grains 

One meta-analysis investigated the protective use of whole grains in the development of cardiovascular diseases and also strokes [30]. This evidence was confirmed even after the dose-response analysis (Appendix A).

### 3.11. Fruit and Vegetables

Aune’s research illustrated the benefit of consumption of fruits and vegetables against the onset of stroke (Table 1, Figure 3). The benefit appears evident in the dose-response study, particularly for certain categories of plant-based foods, such as citrus fruits and citrus juices, for ischemic and hemorrhagic stroke, and the consumption of leafy vegetables for the onset of only ischemic stroke [19] (Appendix A).

### 3.12. Vitamin B Complex

A recent meta-analysis shows that folic acid can reduce stroke risk with an RR = 0.79 (0.68–0.92); while, the combined intake of folic acid and other B-complex vitamins does not appear to be significant, with an RR = 0.91 (0.82–1.00) (Table 1, Figure 3) [27]. 

### 3.13. Carbohydrate Intake

A meta-analysis analyzed the incidence of stroke with respect to the total consumption of carbohydrates as well as glycemic index and glycemic load [44]. The risk of stroke incidence was significant in foods with a higher glycemic load: RR = 1.19 (1.05–1.36). No statistical significance was found for the consumption of the glycemic carbohydrate index (RR= 1.1, 0.99–1.21) and for global carbohydrate consumption (RR = 1.12, 0.93–1.25) (Table 1, Figure 3) [44].

### 3.14. Soy

A meta-analysis investigated soy consumption and analyzed 11 observational studies, including four case-controls and seven cohort studies [45]. The categories with high soy consumption were compared to those with low soy consumption. In the cumulative analysis soy consumption reduced the risk of stroke significantly (RR = 0.82, 0.68–0.99) (Table 1, Figure 3) [45].

### 3.15. Fibers

A meta-analysis by Zhang et al. on fiber consumption highlighted how high fiber intakes are associated with a stroke reduction. In particular, high fiber consumption proved to be protective in ischemic stroke (RR = 0.83, 0.74–0.93), but not in hemorrhagic stroke (RR = 0.87, 0.72–1.05). The dose-response analysis showed that the daily intake of 5 g of fiber leads to a risk reduction (RR = 0.90, 0.82–0.99). A further increase of 10 g shows a higher decrease of RR = 0.84 (0.75–0.94) (Table 1, Figure 3) [46].

### 3.16. Protein

Zhang et al. [47] showed that total protein consumption does not affect stroke risk. However, the consumption of vegetable proteins could be protective (RR = 0.90; 0.82; 0.99) (Table 1, Figure 3).

### 3.17. Fish

Qin’s meta-analysis investigated fish consumption [48]. There is no significant relative risk in the comparison between the consumption of lean fish and fatty fish (RR = 0.88; 0.74–1.04), while there is a protective effect in the consumption of large quantities of lean fish compared to the consumption of small quantities of lean fish (RR = 0.81; 0.67–0.99). Xun’s meta-analysis [49] showed how large consumption of fish has a protective effect against stroke: RR = 0.91 (0.85–0.98) (Table 1, Figure 3).

### 3.18. Meat

Kim et al. investigated the incidence of stroke with respect to meat consumption. Red meat consumption was associated with an increased risk (RR = 1.11; 1.03–1.20). On the other hand, there was a protective effect in the consumption of white meat (RR = 0.87; 0.78–0.96) (Table 1, Figure 3) [50].

### 3.19. Chocolate

Chocolate consumption shows a protective effect against stroke: RR = 0.84 (0.78–0.90) (Table 1, Figure 3) [51].

### 3.20. Flavonoids

High consumption of flavonoids investigated in the meta-analysis by Tang et al. is stroke protective (RR = 0.89; 0.82–0.97). A daily increase of 100 g showed no statistically significant results (RR = 0.91; 0.77–1.08) (Table 1, Figure 3) [52].

### 3.21. Vitamin C

The meta-analysis of Chen et al. concerned vitamin C intake [53]. Consumption of high doses was preventive in the development of ischemic or hemorrhagic stroke (RR = 0.81; 0.74–0.90). Similarly, the dose-response analysis verified that the incremental intake of 100 mg/day of vitamin C has a protective role in the incidence of stroke, RR = 0.82 (0.75–0.93) (Appendix A). In particular, the intake of vitamin C would seem to be protective against ischemic stroke, RR = 0.77 (0.64–0.92), but not hemorrhagic (RR = 1.07; 0.38–3.00) (Table 1, Figure 3, Figure 4, Figure 5 and Figure 6).

### 3.22. Legumes

The consumption of 100 g per week of pulses showed RR = 1.07 (0.77–1.50), with regard to ischemic stroke and RR = 1.23 (0.91; 1.66) as regards to hemorrhagic stroke (Table 1, Figure 3) [54].

### 3.23. Eggs

A moderate consumption of eggs is associated with a potential decrease of stroke, RR = 0.88 (0.81–0.97) (Table 1, Figure 3) [55].

### 3.24. Geographical Distribution of Primary Studies

As regards to geographical distribution of the primary studies, with respect to beverage, food, eating habits or nutrients, there is a strong difference among the areas considered (Figure 2, Table 2). Europe and the USA are areas where the majority of studies were conducted: 162 in Europe (42%) and 130 in the USA (33.7%). It is important to underline that studies about diet style were not conducted in Canada and Australia. 

Eight studies on cereals were conduct in Europe (4.9%), 5 in the USA (3.4%), 3 in Japan (5%), and 1 in China–Singapore–Korea (4%). There was a similar trend for fruits and vegetables: 22 (13.6%) studies in Europe, 14 in the USA (11.6%), 7 in Japan (11.6%), and 1 (4%) in China–Korea–Singapore area.

It is important to underline that Japan followed Europe and the USA in studies pertaining to alcohol use (Figure 2, Table 2); respectively, they have conducted 7 (11.6%), 10 (6.2%), and 13 (5.3%) works respectively, while only 2 studies were done in the China–Korea–Singapore region (9%). All areas considered have studied nutrients (omega-3) with particular attention (Figure 2, Table 2). 

## 4. Discussion

Our review aims to carry out an overview of meta-analyses about the impact of nutrition in the prevention of ischemic/hemorrhagic stroke. Compared to a recent review [56] we wanted to underline some aspects: first, the geographical setting of conducting individual primary studies; second, the study design of the primary studies (observational RCTs); and third, methodological quality of meta-analyses. With respect to the first point, it is important to underline that all primary studies came from countries with high income levels. This evidence shows that many countries are not represented, consequently, different lifestyles, ethnic groups, and potentially harmful or virtuous eating habits are not reported. Moreover, different production standards, regulated by different national or international legislation, could influence the final summary of the data in evidence.

Omega-3 and omega-6 integrators are the most studied, both in meta-analyses of observational studies and RCTs. Discrepancies emerge regarding long-chain omega-3 between the meta-analysis of Larsson [17] and that of Abdelhamid [35]; this difference is likely attributable to a greater sample size in Larsson’s meta-analysis and to more recent publications.

Another highly studied integrator is vitamin C (in China–Singapore–Korea, Europe and the USA). Vitamin C could have a neuroprotective action due to its antioxidant activity. 

However, a Japanese population-based study noted that vitamin C neuroprotection activity would be more effective in non-smokers than smokers, demonstrating that overall lifestyle is responsible for cardiovascular events [57].

Flavonoids act similar to vitamin C. Studies have been conducted in Europe, the USA and China–Singapore–Korea area (Figure 2). Flavonoids perform a neuroprotective action through a triple mechanism: reducing reactive oxygen species (ROS), reducing intracellular concentration of glutamate, and inducing the production of nitric oxide (NO) by activating the enzyme NO-synthase, a powerful vasodilator [58].

The role of some vitamins in relation to cardiovascular risk has also been studied. B vitamins, in particular folic acid, may be linked to the improvement of endothelial function, associated with the increase of 5-methyltetrahydrofolate reductase with the reduction of the circulating homocysteine [59]. Instead vitamin E could play a role in endothelial homeostasis in respect to local inflammation, lipid metabolism, and the stability of atherosclerotic plaques [60].

Comparing the geographical areas examined, the USA and Europe show particular attention to lifestyles. In fact, numerous studies have been conducted in these continents also in relation to alcohol consumption (Figure 2, Table 2). This data could be considered as an indicator of awareness with respect to food education policies and social habits which, however, appear to be very different between different nations, as in the case of Europe [61]. It is well known how the adoption of a healthy diet, with an adequate intake of carbohydrates, greatly reduces cardiovascular risk and obesity [62,63]. With respect to the consumption of soft drinks, it is noted that in Narain’s meta-analysis there is an increased risk for ischemic stroke in women [43]. A recent work by Mullie et al. [64] showed that the daily consumption of soft drinks and similar drinks increases the risk of mortality from cerebrovascular diseases. Regarding tea consumption there are primary studies (Figure 2, Appendix A). Tea as a drink originated in Asia and consumption is widespread worldwide. Among the other substances contained in tea (*Camellia sinensis*) the beneficial effects are attributed mostly to catechins. Catechins are molecules with a positive effect on endothelial function [64]. The benefit of this product for both Asians and Non-Asians was shown in a meta-analysis by Arab et al. [26]. There are many studies on cereals in a large part of the areas considered (Figure 2). It is important to underline that the consumption of fresh fruit, nuts, and legumes entails a potential risk reduction [19,21,45,54].

Their consumption is encouraged by all the most recent guidelines on cardiovascular prevention [56,65,66] even though there are notable differences between geographical areas and social context [21,66]. As pointed out by Lake et al., climate change could also affect the accentuation of inequalities in access to food and healthy food, particularly in developing countries [67,68].

The results of the studies regarding red meat are controversial. Excessive consumption of red meat and specially processed meat, studied in only two geographical areas (Europe and the USA), show an increase in risk; while moderate consumption of red meat does not lead to an alteration of the lipid structure or a significant pressure rise [69]. Moreover, cardiovascular risk could be mitigated by the adequate consumption of fruit and vegetables [70,71]. 

Finally, it is important to underline that some widespread types of cancer, such as colorectal and breast cancer [69,70,71], have many risk factors in common with cardiovascular diseases.

Particular importance is the intestinal microbiome. Some studies suggest that dysbiosis may favor ischemic stroke. A study by Yin et al. showed that the bacterial flora of patients with stroke was rich in some opportunistic bacteria (*Enterobacter*, *Megasphaera*, *Oscillibacter,* and *Desulfobivrio*) compared to saprophytic flora (*Bacterioides, Prevotella* and *Faecalibacterium*) [72]. A work by Xia et al. showed a substantial difference in the microbiome between ischemic patients and control subjects [73].

A limitation of the present study is related to the design of the study of primary studies. In fact, the basal conditions and the possible comorbidities of the subjects enrolled in these studies are not known.

## 5. Conclusions

Most physicians and health professionals underestimate the importance of food and lifestyles, smoking, consumption of alcohol, and daily exercise as stroke risk factor. It is very important to underline nutrition in stroke prevention.

This review reveals that choosing foods with a more favorable nutritional profile may help reduce the risk of cardiovascular diseases and stroke in particular. These indications can be specifically addressed to those classes of the population with an increased risk of stroke, using a “tailored” preventive medicine for individuals based on genetic predisposition, presence of other risk factors or predisposing lifestyles.

Although far from identifying a “superfood” with nutraceutical properties that can guarantee absolute well-being or zero risk, it is clear that the choice of a balanced diet can reduce the risk of stroke, a disease with high social costs.

In the nineteenth century, Ludwig Feuerbach wrote “You are what you eat”. The research carried out so far on nutrition confirms this brilliant statement. Governments should back public health policies and promote healthy lifestyles. 

## Figures and Tables

**Figure 1 ijerph-16-03582-f001:**
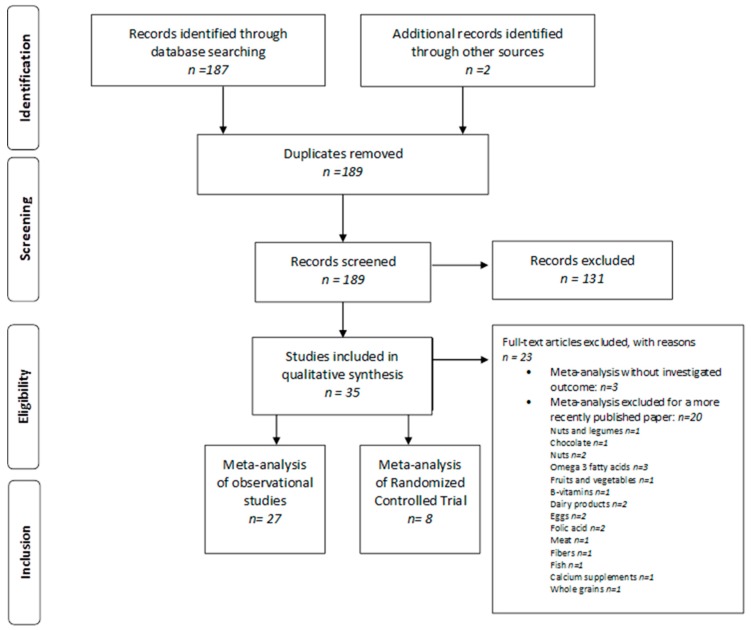
The preferred reporting items for systematic reviews and meta-analyses (PRISMA) flow chart.

**Figure 2 ijerph-16-03582-f002:**
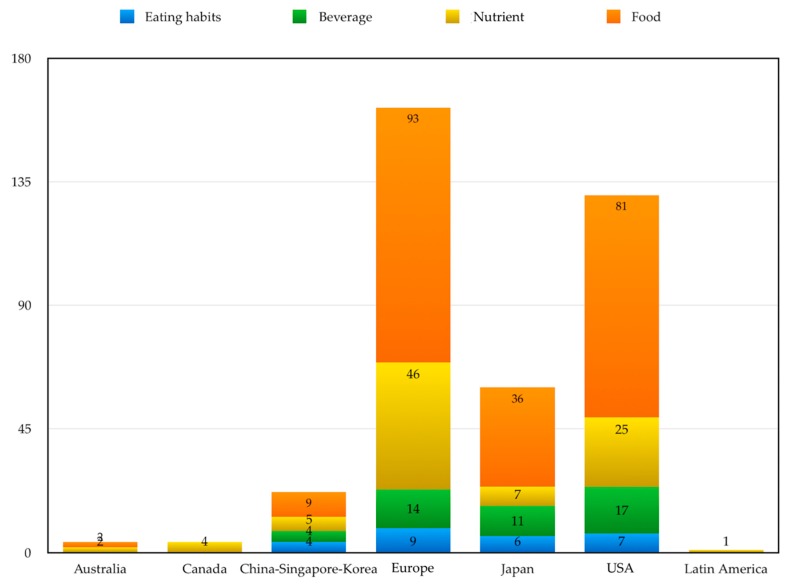
Distribution of primary studies included in meta-analyses considered according to geographic area and type of nutritional support.

**Figure 3 ijerph-16-03582-f003:**
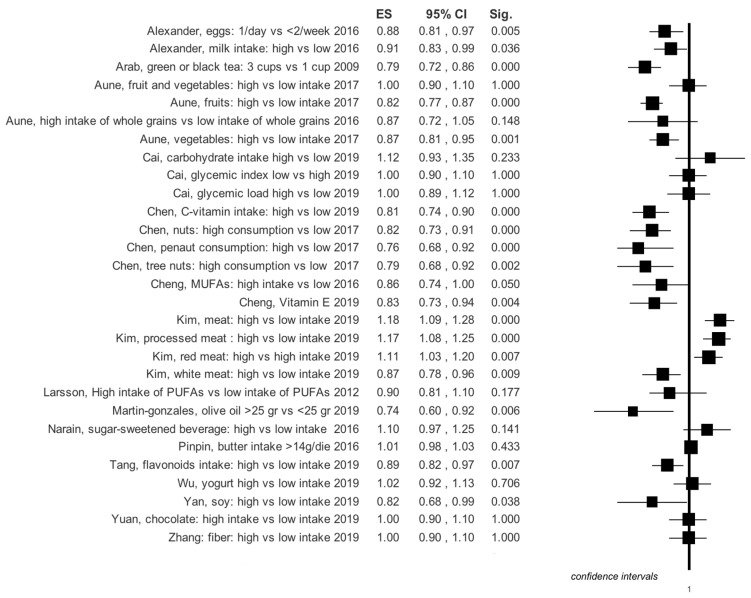
Summary results of effects size for any type of stroke events in observational studies, based on study design of selected primary studies for each meta-analysis.

**Figure 4 ijerph-16-03582-f004:**
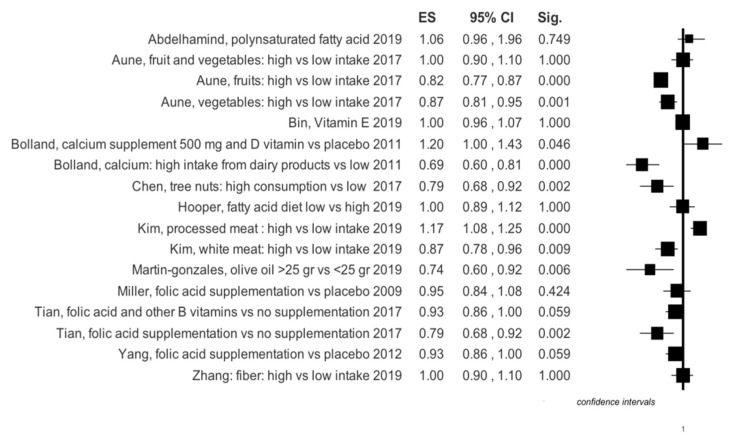
Summary results of effects size for any type of stroke events in RCT, based on study design of selected primary studies for each meta-analysis.

**Figure 5 ijerph-16-03582-f005:**
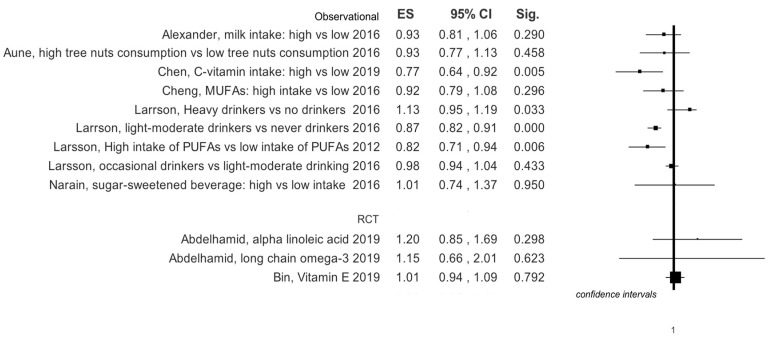
Summary results of effects size for ischemic of stroke events, based on study design of selected primary studies for each meta-analysis.

**Figure 6 ijerph-16-03582-f006:**
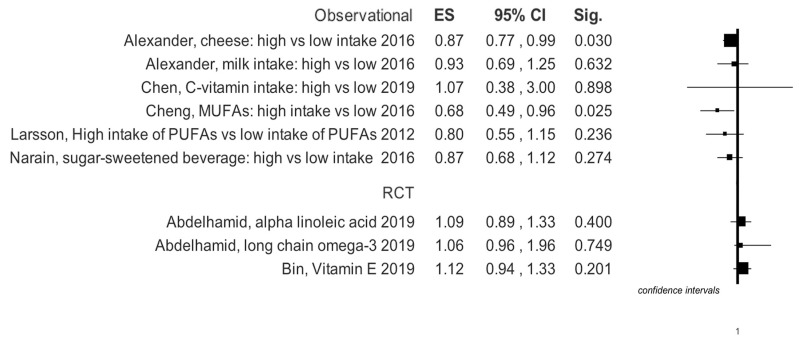
Summary results of effects size for hemorrhagic of stroke events, based on study design of selected primary studies for each meta-analysis.

**Table 1 ijerph-16-03582-t001:** Characteristics of included meta-analyses in the overview according to food or beverage, study design, and type of stroke.

Author	Food or Beverage	Control Group	Literature Search Update	Number of Primary Studies	Type of Strokes *	Number of Studies for Evaluated Strokes	Population		Effect Size 95% CI
							No. Total	No. Events	
Observational Studies									
Alexander [5]	High milk intake	Low milk intake	2016	K = 31	Ischemic or Hemorrhagic	7	-	-	0.91 (0.83; 0.99)
	High milk intake	Low milk intake			Ischemic or Hemorrhagic (in men)	4	-	-	1.04 (0.96; 1.14)
	High milk intake	Low milk intake			Ischemic or Hemorrhagic	4	-	-	0.93 (0.81; 1.06)
	High milk intake	Low milk intake			Hemorrhagic	3	-	-	0.93 (0.69; 1.25)
	High cheese intake	Low cheese intake			Ischemic or Hemorrhagic	4	-	-	0.87 (0.77; 0.99)
Zhang [7]	Western dietary pattern—high categories #	Western dietary pattern—low categories #	2015	K = 21	Ischemic or Hemorrhagic	8	143,798	2049	1.05 (0.82; 1.35) **
	Healthy dietary pattern—high categories #	Healthy dietary pattern—low categories #				14	318,813	3971	0.77 (0.64; 0.93) **
Mullie [8]	200 mL/day daily milk consumption	No milk consumption	2016	K = 19	Ischemic or Hemorrhagic	10	567,717	39,352	0.91 (0.82; 1.02)
					Ischemic or Hemorrhagic (in men)	5	*-*	*-*	0.96 (0.86; 1.09)
Pimpin [9]	Butter intake <14 g/day	Butter intake >14 g/day	2016	K = 4	Ischemic or Hemorrhagic	3	173,853	5229	1.01 (0.98; 1.03)
Larsson [12]	Light-moderate drinking	No drinkers	2016	K = 27	Ischemic stroke	8	-	-	0.87 (0.81; 0.92)
	Heavy drinking	No drinkers				8	-	-	1.13 (0.95; 1.19)
	Light-moderate drinking	Never drinkers				8	-	-	0.87 (0.82; 0.91)
	Heavy drinking	Never drinkers				8	-	-	1.06 (0.95; 1.19)
	Occasional drinking	Light-moderate drinkers				8	-	-	0.98 (0.94; 1.04)
	Heavy drinking	Occasional drinkers				8	-	-	1.13 (1.03; 1.24)
	Light-moderate drinking	No drinkers			Intracerebral hemorrhage	5	-	-	0.91 (0.64; 1.29)
	Heavy drinking	No drinkers				4	-	-	1.21 (0.87; 1.67)
	Light-moderate drinking	Occasional drinkers				4	-	-	1.04 (0.89; 1.21)
	Heavy drinking	Occasional drinkers				4	-	-	1.74 (1.45; 2.09)
	Light-moderate drinking	No drinkers			Subarachnoid Hemorrhage events	5	-	-	1.39 (1.00; 1.92)
	Heavy drinking	No drinkers				3	-	-	1.43 (1.00; 2.05)
	Light-moderate drinking	Occasional drinkers				4	-	-	1.10 (0.84; 1.44)
	Heavy drinking	Occasional drinkers				4	-	-	1.62 (0.89; 2.29)
Cheng [16]	High monounsaturated fatty acids (MUFAs) intake	Low usage of MUFAs	2016	K = 10	Ischemic or Hemorrhagic	10	314,511	5827	0.86 (0.74; 1.00)
					Ischemic stroke	8		-	0.92 (0.79; 1.08)
					Hemorrhagic stroke	5	-	-	0.68 (0.49; 0.96)
Larsson [17]	High long-chain omega-3 polyunsaturated fatty acids (PUFAs) intake	Low intake of PUFA	2012	K = 10	Ischemic or Hemorrhagic	10	242,076	5238	0.90 (0.81; 1.10)
					Ischemic stroke	5	-	-	0.82 (0.71; 0.94)
					Hemorrhagic stroke	5	-	-	0.80 (0.55; 1.15)
Martin-Gonzales [40]	Olive oil (>25 g)	Olive oil (<25 g)	2014	K = 2	Ischemic or Hemorrhagic	2	*-*	*-*	0.74 (0.60; 0.92)
Cheng [41]	Vitamin E	-	2018	K = 9	Ischemic or Hemorrhagic	9	-	-	0.83 (0.73; 0.94)
Aune [19]	High intake of fruit and vegetables	Low intake of fruit and vegetables	2017	K = 95	Ischemic or Hemorrhagic	8	226,910	10,560	0.79 (0.71; 0.88)
	High intake of fruit	Low intake of fruit			Ischemic or Hemorrhagic	17	960,337	46,951	0.82 (0.77; 0.87)
	High intake vegetables	Low intake vegetables			Ischemic or Hemorrhagic	13	427,124	14,519	0.87 (0.81; 0.95)
	High intake apples and pears	Low intake apples and pears			Ischemic or Hemorrhagic	6	-	-	0.88 (0.81; 0.96)
	High intake berries	Low intake berries			Ischemic or Hemorrhagic	5	-	-	0.98 (0.86; 1.12)
	High intake citrus fruits	Low intake citrus fruits			Ischemic or Hemorrhagic	8	-	-	0.74 (0.65; 0.84)
	High intake citrus fruit juice	Low intake citrus fruit juice			Ischemic or Hemorrhagic	2	-	-	0.90 (0.74; 1.10)
	High intake dried fruits	Low intake dried fruits			Ischemic or Hemorrhagic	2	-	-	0.92 (0.74; 1.15)
	High intake fruits juice	Low intake fruit juice			Ischemic or Hemorrhagic	2	-	-	0.67 (0.60; 0.76)
	High intake grapes	Low intake grapes			Ischemic or Hemorrhagic	2	-	-	0.72 (0.47; 1.10)
	High intake allium vegetables	Low intake allium vegetables			Ischemic or Hemorrhagic	2	-	-	0.89 (0.80; 1.00)
	High intake cruciferous vegetables	Low intake cruciferous vegetables			Ischemic or Hemorrhagic	4	-	-	0.97 (0.78; 1.20)
	High intake green leafy vegetables	Low intake green leafy vegetables			Ischemic or Hemorrhagic	4	-	-	0.88 (0.81; 0.95)
	High intake pickled vegetables	Low intake pickled vegetables			Ischemic or Hemorrhagic	2	-	-	0.80 (0.73; 0.88)
	High intake potatoes	Low intake potatoes			Ischemic or Hemorrhagic	4	-	-	0.94 (0.87; 1.01)
	High intake root vegetables	Low intake root vegetables			Ischemic or Hemorrhagic	2	-	-	1.01 (0.89; 1.14)
	High intake tomatoes	Low intake tomatoes			Ischemic or Hemorrhagic	3	-	-	0.95 (0.68; 1.31)
	High intake berries	Low intake berries			Ischemic	3	-	-	0.95 (0.75; 1.21)
	High intake citrus fruits	Low intake citrus fruits			Ischemic	7	-	-	0.78 (0.66; 0.92)
	High intake citrus fruit juice	Low intake citrus fruit juice			Ischemic	2	-	-	0.65 (0.51; 0.84)
	High intake allium vegetables	Low intake allium vegetables			Ischemic	2	-	-	0.90 (0.78; 1.03)
	High intake cruciferous vegetables	Low intake cruciferous vegetables			Ischemic	5	-	-	0.82 (0.66; 1.01)
	High intake green leafy vegetables	Low intake green leafy vegetables			Ischemic	4	-	-	0.88 (0.78; 0.99)
	High intake potatoes	Low intake potatoes			Ischemic	5	-	-	0.97 (0.87; 1.08)
	High intake root vegetables	Low intake root vegetables			Ischemic	3	-	-	0.93 (0.73; 1.18)
	High intake tomatoes	Low intake tomatoes			Ischemic	2	-	-	0.80 (0.69; 0.92)
	High intake berries	Low intake berries			Hemorrhagic	3	-	-	1.15 (0.89; 1.49)
	High intake citrus fruits	Low intake citrus fruits			Hemorrhagic	3	-	-	0.74 (0.55; 1.01)
	High intake cruciferous vegetables	Low intake cruciferous vegetables			Hemorrhagic	2	-	-	0.83 (0.33; 2.12)
	High intake potatoes	Low intake potatoes			Hemorrhagic stroke	3	-	-	1.06 (0.83; 1.36)
	High intake root vegetables	Low intake root vegetables			Hemorrhagic stroke	2	-	-	1.05 (0.76; 1.44)
Chen [21]	All nuts high consumption	All nuts low consumption	2017	K = 16	Ischemic or Hemorrhagic	12	449,293	4398	0.82 (0.73; 0.91)
	Nut plus peanut butter high consumption	Nut plus peanut butter low consumption				3	104,531	924	0.84 (0.70; 1.01)
	Peanuts high consumption	Peanuts low consumption				5	265,252	7025	0.76 (0.69; 0.82)
	Tree nuts high consumption	Tree nuts low consumption				3	130,987	6394	0.79 (0.68; 0.92)
Aune [30]	High intake of whole grains or specific types of grains	Low intake of whole grains or specific types of grains	2016	K = 15	Ischemic or Hemorrhagic	5	-	-	0.87 (0.72; 1.05)
	High intake whole grain bread	Low intake whole grain bread				2	-	-	0.88 (0.75; 1.03)
	High intake of whole grain breakfast cereals	Low intake of whole grain breakfast cereals				2	-	-	0.99 (0.53; 1.86)
	High intake of refined grain	Low intake of refined grain				4	-	-	0.95 (0.78; 1.14)
	High intake total rice	Low intake total rice				4	-	-	1.02 (0.94; 1.11)
Wu [33]	High yogurt intake	Low yogurt intake	2017	K = 7	Ischemic or Hemorrhagic	7	*-*	*-*	1.02 (0.92; 1.13)
Muto [38]	High saturated fatty acid intake	Low saturated fatty acid intake	2018	K = 16	Ischemic	11	*-*	*-*	0.88 (0.81; 0.96)
Narain [43]	High intake sugar-sweetened beverages	Low intake sugar-sweetened beverages	2016	K = 7	Ischemic or Hemorrhagic	3	236,061	-	1.10 (0.97; 1.25)
	High intake sugar-sweetened beverages	Low intake sugar-sweetened beverages			Ischemic stroke (in men)	3	-	-	1.01 (0.74; 1.37)
	High intake sugar-sweetened beverages	Low intake sugar-sweetened beverages			Ischemic stroke (in women)	3	-	-	1.33 (1.07; 1.66)
	High intake sugar-sweetened beverages	Low intake sugar-sweetened beverages			Hemorrhagic stroke (in men)	3	-	-	0.87 (0.68; 1.12)
	High intake sugar-sweetened beverages	Low intake sugar-sweetened beverages			Hemorrhagic stroke (in women)	3	-	-	0.83 (0.62; 1.10)
Cai [44]	Glycemic index	-	2014	K = 7	Ischemic or Hemorrhagic	7	-	-	1.10 (0.99; 1.21)
	Glycemic load								1.19 (1.05; 1.36)
	Carbohydrate intake								1.12 (0.93; 1.35)
Yan [45]	High soy consumption	Low soy consumption	2016	K = 11	Ischemic or Hemorrhagic	11	*-*	*-*	0.82 (0.68; 0.99)
Zhang [46]	High fiber intake	Low fiber intake	2013	K = 11	Ischemic or Hemorrhagic	11	325,627	*-*	0.83 (0.74; 0.93)
					Ischemic	8	*-*	*-*	0.83 (0.74; 0.93)
					Hemorrhagic	5	*-*	*-*	0.87 (0.74; 1.05)
Zhang [47]	Protein intake	-	2016	K = 12	Ischemic or Hemorrhagic	12	*-*	*-*	0.98 (0.89; 1.07)
					Ischemic	8			0.94 (0.80; 1.10)
					Hemorrhagic	4			1.05 (0.97; 1.14)
	Animal protein	-			Ischemic or Hemorrhagic	8			0.94 (0.75; 1.17)
	Vegetable protein	-			Ischemic or Hemorrhagic	8			0.90 (0.82; 0.99)
Qin [48]	Lean fish	Fatty fish	2018	K = 5	Ischemic or Hemorrhagic	5	*-*	*-*	0.88 (0.74; 1.04)
	High lean fish intake	Low lean fish intake	2018	K = 5	Ischemic or Hemorrhagic	5	*-*	*-*	0.81 (0.67; 0.99)
Xun [49]	High fish intake	Low fish intake	2012	K = 16	Ischemic or Hemorrhagic	16	*-*	*-*	0.91 (0.85; 0.98) *
Kim [50]	High total meat intake	Low total meat intake	2016	K= 7	Ischemic or Hemorrhagic	6	*-*	*-*	1.18 (1.09; 1.28)
	High red meat intake	Low red meat intake				7	*-*	*-*	1.11 (1.03; 1.20)
	High processed meat intake	Low processed meat intake				8	*-*	*-*	1.17 (1.08; 1.25)
	High white meat intake	Low white meat intake				4	*-*	*-*	0.87 (0.78; 0.96)
Yuan [51]	High chocolate intake	Low chocolate intake	2017	K = 8	Ischemic or Hemorrhagic	8	*-*	*-*	0.84 (0.78; 0.90)
Tang [52]	High flavonoids intake	Low flavonoids intake	2016	K = 11	Ischemic or Hemorrhagic	11	-		0.89 (0.82; 0.97)
Chen [53]	High vitamin C intake	Low vitamin C intake	2011	K = 11	Ischemic or Hemorrhagic	11	*-*	*-*	0.81 (0.74; 0.90)
					Ischemic	4			0.77 (0.64; 0.92)
					Hemorrhagic	2	*-*	*-*	1.07 (0.38; 3.00)
Afshin [54]	Legumes 100 g/week	No consumption	2014	K = 6	Ischemic	3	-	-	1.07 (0.77; 1.50),
					Hemorrhagic	4	-	-	1.23 (0.91; 1.66)
Alexander [55]	1 egg/day	<2 eggs/week	2016	K = 7	Ischemic or Hemorrhagic	7	*-*	*-*	0.88 (0.81; 0.97)
RCT									
Bolland [34]	High Ca from dairy products	Low Ca from dairy products	2011	K = 8	Ischemic or Hemorrhagic	5	-	-	0.69 (0.60; 0.81)
	Calcium supplement 500 mg and D vitamin	Placebo			Ischemic or Hemorrhagic	3	20,090	477	1.20 (1.00; 1.43)
Tian [27]	Intervention regimen folic acid (FA) ## only	No supplementation	2017	K = 11	Ischemic or Hemorrhagic	11	21,295	657	0.79 (0.68; 0.92)
	Intervention regimen FA + vitamin B	No supplementation					27,486	1589	0.91 (0.82; 1.00)
Arab [26]	Tea 3 cups	Tea 1 cup	2009	K = 9	Ischemic or Hemorrhagic	9	*-*	*-*	0.77 0.71; 0.85
Abdelhamid [35]	High long-chain omega-3 polyunsaturated fatty acids (PUFAs) intake	Low PUFAs intake	2018	K = 32	Ischemic or Hemorrhagic	28	89,358	1818	1.06 (0.96–1.16)
	High alpha linoleic acid intake	Low alpha linoleic acid intake			Ischemic or Hemorrhagic	4	19,327	51	1.15 (0.66; 2.01)
Hooper [36]	Low omega-6	High omega-6 intake	2018	K = 4	Ischemic or Hemorrhagic	4	3730	54	1.36 (0.45; 4.11)
Abdelhamid [37]	High polyunsaturated fatty acid intake	Low polyunsaturated fatty acid intake	2018	K = 11	Ischemic or Hemorrhagic	11	14,724	165	1.06 (0.96; 1.96)
Hooper [39]	Low saturated fatty acid diet	Low saturated fatty acid diet	2015	K = 8	Ischemic or Hemorrhagic	8	50,952	1125	1.00 (0.89; 1.12)
Bin [42]	Vitamin E	-	2011	K = 13	Ischemic or Hemorrhagic	13	166,282	-	1.01 (0.96; 1.07)
					Ischemic	-	-	-	1.01 (0.94; 1.09)
					Hemorrhagic	-	-	-	1.12 (0.94; 1.33)

* Where not specified, stroke events is in both sexes. # Dietary pattern: high intake of all kinds of red and/or processed meats, refined grains, sweets, desserts, high-fat dairy products, and high-fat gravy. ** OR (odds ratio). ## Folic acid.

**Table 2 ijerph-16-03582-t002:** Distribution of primary studies included in meta-analyses considered according to geographic area and type of nutritional support.

Total*n* = 386	Australia*n* = 6 (1.5%)	Canada*n* = 4 (1%)	China-Singapore-Korea*n* = 22 (5.7%)	Europe*n* = 162 (42%)	Japan*n* = 60 (15.5%)	USA*n* = 130 (33.67%)	Latin America*n* = 1 (0.25%)
*n*	%	*n*	%	*n*	%	*n*	%	*n*	%	*n*	%	*n*	%
Eating habits														
Healthy diet					4	19	5	3.1	6	10	1	0.77		
Carbohydrates							4	2.4			6	4.6		
Beverages														
Alcohol					2	9	10	6.2	7	11.6	7	5.3		
Tea	1	17			1	9	3	1.8	3	5	7	5.3		
Soft drinks					1	9	1	0.6	1	1.6	3	2.3		
Nutrients								0				0		
Omega-3	1	17	1	0.15	1	4	4	2.5	3	5	4	3.0	1	100
Folic acid			3	0.75	1	4	9	5.5	1	1.6	1	0.7		
Monounsaturated fatty acids	1	17					3	1.8	2	3.3	4	3.1		
Polyunsaturated fatty acids							3	1.8			2	1.5		
Omega-6							2	1.2			1	0.7		
Flavonoids					1	4	6	3.7			4	3.0		
Vitamin E					1	4	11	6.8	3	5	6	4.6		
Vitamin C					1	4	6	3.7			3	2.3		
Calcium/vitamin D							2	1.2			1	0.7		
Food												0		
Dried fruits	2	34			1	4	3	1.8		0	2	1.5		
Saturated fatty acids							2	1.2	5	8.3	2	1.5		
Butter							3	1.8		0	5	3.8		
Meat							4	2.5		0	1	0.7		
Cereals					1	4	8	4.9	3	5	1	0.7		
Chocolate							5	3.1	1	1.6	4	3.1		
Fibers					1	14	3	1.8	2	3.3	14	10.7		
Fruits and vegetables					3		22	13.6	7	11.6	11	8.5		
Milk							6	3.7	2	3.3	5	3.9		
Milk and derivatives							5	3.1	3	5	2	1.5		
Legumes							2	1.2	2	3.3	8	6.1		
Olive oil							2	1.2		0				
Fish					1	4	12	7.4	3	5	6	4.6		
Protein					1	4	2	1.2	3	5	2	1.5		
Soy					1	4	2	1.2	2	3.3	5	3.8		
Yogurt							7	4.3		0	13	10		
Eggs							5	3.1	1	1.6	2	1.5

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
