# Peer review of "Overview of Meta-Analyses: The Impact of Dietary Lifestyle on Stroke Risk"

_ijerph, 2019, doi:10.3390/ijerph16193582_

Round 1

Reviewer 1 Report

The authors summarize current meta-analyses on nutrition and stroke risk, but as they recognise, a recent work by Iacoviello et al. did the same. Authors mark that their novelty is the consideration of the geographic origins of the primary works.

General observations:  It is necessary to check if any study is present in more than one meta-analyses of the same topic. Most of the affirmations are based only in one meta-analysis. Please, double check bibliography, as there are too many wrong references. 

Novelty of current work is to include geographic origin of the studies, in my opinion, authors should develop more in deep this point.  It is interesting to know if different nutrients have different influence on stroke depending on geographic area. If stroke risk factors are the same between the different regions analysed, and if the nutrition habits of the regions analysed are the same.  

Lines 163 to 244 are merely a comment on different papers. If these data and papers are present in tables and figures, please mention it.

Line 42: Reference 9 is a study about milk, not alcohol.

Line 42: Reference 10 is a study about butter, not alcohol.

Lines 42-44, please add reference about wine properties

Line 45 and 46: References 12 and 13 are also wrong

Lines 47-48, References 14 and 15 are not about body fatty acid modification, but about effects of MUFA and PUFA consumption on stroke. Moreover, reference 16 reports this effect only for MUFA. Please, remove wrong references and rephrase correctly.

Line 56: Reference 22 is wrong

Line 60: Reference 23 is wrong

Figure 2: Add legend for colours used in bars. Move to results, as the authors emphasize that is the novelty of the paper.  Missing data for Blue zones

Tables 1, 2 and 3 should go to supplementary data. Tables 1 and 2 are Results.

Line 198: Not the correct reference

Lines 199-209 format

Line 227 format

Line 172: Reference 17 is wrong

Lines 250-260: Put data in context. Which is the % of studies in each countries that are about Stoke and Nutrition? In each geographic area, which is the % of studies for each nutrient? That is important, as authors say that Europe and USA have a “particular interest to investigate alcohol use”, but data from this work indicates that in Europe only 6.2% of the studies are about stroke and alcohol, meanwhile data for USA and China is of 10.4%. Moreover, this affirmation is based on the papers present on the selection of the meta-analyses they performed. That means that there are two biases present, the first is on the studies selected on each meta-analyses and the second on the meta-analyses selection that authors performed.

Line 277-278: Current work is about the impact of lifestyle on stroke risk, but reference 55 is about reducing ischemic damage, please remove it.

Line 286-287: Repeated sentence (lines 279-280).

Lines 300-302: Camellia sinensis (that’s the correct spelling of the name) is indeed the tea plant, and its leaves contain the catechins. Please, rephrase.

Lines 306 and 304-314 are about the same, put it together.

Lines 316-317: This sentence is not necessary, and is not adding any value to the work but including three self-citations. Please remove it.

Lines 330-331: Is 1850 the correct date of Feuerbach’s phrase? Could be 1860-1864?  

Author Response

Dear Editor,

Enclosed please find the manuscript formerly entitled “ Overview of meta-analyses: the impact of dietary lifestyle on stroke risk” by Altobelli E, Paolo Matteo Angeletti, Leonardo Rapacchietta, Reimondo Petrocelli.

We have addressed all the comments and suggestions of reviewers 1, 2 and 3.

We have highlight in yellow all changes.

Title. We modified tile in: Overview of meta-analyses: the impact of dietary lifestyle on stroke risk.

Answers to Reviewer

Reviewer 1

The authors summarize current meta-analyses on nutrition and stroke risk, but as they recognise, a recent work by Iacoviello et al. did the same. Authors mark that their novelty is the consideration of the geographic origins of the primary works.

General observations:  It is necessary to check if any study is present in more than one meta-analyses of the same topic. Most of the affirmations are based only in one meta-analysis.

We added the following sentence: To avoid redundant results, we only considered the last published meta-analysis for the same-topic (lines 84-85).

Please, double check bibliography, as there are too many wrong references. 

We thank the reviewer. We corrected wrong references.

Novelty of current work is to include geographic origin of the studies, in my opinion, authors should develop more in deep this point.  It is interesting to know if different nutrients have different influence on stroke depending on geographic area. If stroke risk factors are the same between the different regions analysed, and if the nutrition habits of the regions analysed are the same.  

We added some sentences in the discussion section (lines 368-370).

Lines 163 to 244 are merely a comment on different papers. If these data and papers are present in tables and figures, please mention it.

We have included tables and figures in the main text.

Line 42: Reference 9 is a study about milk, not alcohol.

Line 42: Reference 10 is a study about butter, not alcohol.

Lines 42-44, please add reference about wine properties

Line 45 and 46: References 12 and 13 are also wrong

Lines 47-48, References 14 and 15 are not about body fatty acid modification, but about effects of MUFA and PUFA consumption on stroke. Moreover, reference 16 reports this effect only for MUFA. Please, remove wrong references and rephrase correctly.

Line 56: Reference 22 is wrong

Line 60: Reference 23 is wrong

We thank the reviewer. We corrected all references.

Figure 2: Add legend for colours used in bars. Move to results, as the authors emphasize that is the novelty of the paper.  Missing data for Blue zones

Tables 1, 2 and 3 should go to supplementary data. Tables 1 and 2 are Results.

We added legend in figure 2. We changed table 2 in supplementary table 2.

We moved table1 and table 2 (before table 3) and figures 1-6 in the results section.

Line 198: Not the correct reference

We thank the reviewer. We corrected it.

Lines 199-209 format

Line 227 format

We modified format (now 245-248 and 250-255 lines).

Line 172: Reference 17 is wrong

We thank the reviewer. We corrected it.

Lines 250-260: Put data in context. Which is the % of studies in each countries that are about Stoke and Nutrition? In each geographic area, which is the % of studies for each nutrient? That is important, as authors say that Europe and USA have a “particular interest to investigate alcohol use”, but data from this work indicates that in Europe only 6.2% of the studies are about stroke and alcohol, meanwhile data for USA and China is of 10.4%. Moreover, this affirmation is based on the papers present on the selection of the meta-analyses they performed. That means that there are two biases present, the first is on the studies selected on each meta-analyses and the second on the meta-analyses selection that authors performed.

In agreement with referee we added % of studies for each country in the main text and in table 2.

Line 277-278: Current work is about the impact of lifestyle on stroke risk, but reference 55 is about reducing ischemic damage, please remove it.

We removed the sentence and its reference.

Line 286-287: Repeated sentence (lines 279-280).

We deleted sentence.

Lines 300-302: Camellia sinensis (that’s the correct spelling of the name) is indeed the tea plant, and its leaves contain the catechins. Please, rephrase.

We corrected the name.

Lines 306 and 304-314 are about the same, put it together.

We modified the sentences as follows: It is important to underline that the consumption of fresh fruit, nuts and legumes entails a potential risk reduction (line 348-350).

Lines 316-317: This sentence is not necessary, and is not adding any value to the work but including three self-citations. Please remove it.

We removed the sentence and old references 70 and 71.

Lines 330-331: Is 1850 the correct date of Feuerbach’s phrase? Could be 1860-1864?  

We modified 1850 years in XIX century. (we used century because a certain data is not known). (line 383).

We hope that the revised manuscript may be acceptable for publication in International Journal of Environmental Research and Public Health.

Sincerely   

Emma Altobelli

Reviewer 2 Report

This review article by Altobelli et al. focused on relationship between lifestyle, foods, drinks, alcohol, smoking, exercise) and the risk of stroke events (ischemic/hemorrhagic). Authors performed meta-analysis using 27 observational studies and 8 RCT and demonstrated that choosing foods with a more favorable nutritional profile may help to reduce the risk of cardiovascular diseases and stroke in particular. As authors mentioned, the importance of food and lifestyles as stroke risk factor is underestimated by physicians and health professionals. Therefore, the concept of this meta-analysis is valuable and the results seem agreeable. Although overall manuscript seems written very well. Authors may want to resolve several issues as below.

Major comments;

1) Atrial fibrillation (AF) is a strong and independent risk factor for cardiogenic thromboembolism that causes ischemic stroke. In this analysis, were patients with AF included in subjects? And, was cardiogenic stroke included in ischemic stroke?
Authors may want to state it clearly.

Minor comments;

1) Please remake figures 2 and 3 with larger letters.

2) Supplementary Table 1 cannot be found.

Author Response

Dear Editor,

Enclosed please find the manuscript formerly entitled “ Overview of meta-analyses: the impact of dietary lifestyle on stroke risk” by Altobelli E, Paolo Matteo Angeletti, Leonardo Rapacchietta, Reimondo Petrocelli.

We have addressed all the comments and suggestions of reviewers 1, 2 and 3.

We have highlight in yellow all changes.

Title. We modified tile in: Overview of meta-analyses: the impact of dietary lifestyle on stroke risk.

Answers to Reviewer

Reviewer 2

We thank very much the Reviewer for the positive comments.

1) Atrial fibrillation (AF) is a strong and independent risk factor for cardiogenic thromboembolism that causes ischemic stroke. In this analysis, were patients with AF included in subjects? And, was cardiogenic stroke included in ischemic stroke?

Authors may want to state it clearly.

We added the following sentences at the end of the discussion: “A limitation of the present study is related to the design of the study of primary studies. In fact, the basal conditions and the possible comorbidities of the subjects enrolled in these studies are not known”.

Minor comments;

  • Please remake figures 2 and 3 with larger letters.

We modified figures 2 and 3.

2) Supplementary Table 1 cannot be found.

We included table 1S at the end of main text.

We hope that the revised manuscript may be acceptable for publication in International Journal of Environmental Research and Public Health.

Sincerely   

Emma Altobelli

Reviewer 3 Report

Altobelli E. et al. performed an objective meta-analyses study to overview the associations between lifestyle and ischemic/hemorrhagic stroke. The authors included papers from the last 10 years in the MEDLINE, EMBASE, Scopus, Clinicaltrail.gov, Web of Science, and Cochrane Library databases, and selected targeted studies by PRISMA method. 27 meta-analyses on observational studies and 8 randomized control trials were included for the meta-analyses in this manuscript. The authors evaluated the geographical areas of those primary studies and summarized the results by effect size for different types of stroke events among food/nutrients comparisons. It is nice to start from the geographical area since diet habits usually differ from the local weather, agriculture types, and lifestyle. However, I still have several suggestions:

1) Several modifiable factors of lifestyles have been known potentially associated with stroke risk, such as physical activity, eating habits, and smoking. I would suggest to either add analyses/paragraphs of physical activity and smoking or change the title to “the impact of dietary lifestyle on stroke risk”.

2) In the Result section, the authors may want to arrange similar food types and related nutrients into one paragraph (e.g., Olive oil, MUFAs, and PUFAs).

3) Chocolate at line 228 supposes not part of the 3.17. Meat.

4) In 3.23 at line 252, do you mean there is no diet style study among your analyses (35 studies)?

5) Please label the x-axis of forest plots for including the upper and lower limits of 95% CI in each study (Figure 3-5).

6) If possible, discuss some potential effects or biological relevance of microbiome on the impact of diet on stroke.

7) Please provide “supplemental Table 1” as shown in Line 99.

Author Response

Dear Editor,

Enclosed please find the manuscript formerly entitled “ Overview of meta-analyses: the impact of dietary lifestyle on stroke risk” by Altobelli E, Paolo Matteo Angeletti, Leonardo Rapacchietta, Reimondo Petrocelli.

We have addressed all the comments and suggestions of reviewers 1, 2 and 3.

We have highlight in yellow all changes.

Title. We modified tile in: Overview of meta-analyses: the impact of dietary lifestyle on stroke risk.

Answers to Reviewer

Reviewer 3

We thank very much the Reviewer for the positive comments.

  • Several modifiable factors of lifestyles have been known potentially associated with stroke risk, such as physical activity, eating habits, and smoking. I would suggest to either add analyses/paragraphs of physical activity and smoking or change the title to “the impact of dietary lifestyle on stroke risk”.

We changed the title as follows: “Overview of meta-analyses: the impact of dietary lifestyle on stroke risk”.

  • In the Result section, the authors may want to arrange similar food types and related nutrients into one paragraph (e.g., Olive oil, MUFAs, and PUFAs).

We modified consecution of sentences.

  • Chocolate at line 228 supposes not part of the 3.17. Meat.

We separated chocolate text by meat paragraph.

In 3.23 at line 252, do you mean there is no diet style study among your analyses (35 studies)?

We modified the sentence as follows: It is important to underline that studies about diet style were not conducted in Canada and Australia (line 256).

  • Please label the x-axis of forest plots for including the upper and lower limits of 95% CI in each study

We added the words: “confidence interval”.

  • If possible, discuss some potential effects or biological relevance of microbiome on the impact of diet on stroke.

We discuss some potential effects of microbioma on the impact of diet on stroke (lines 362-367).

  • Please provide “supplemental Table 1” as shown in Line 99.

We put “supplement table 1” at the end of main text.

We hope that the revised manuscript may be acceptable for publication in International Journal of Environmental Research and Public Health.

Sincerely   

Emma Altobelli

Round 2

Reviewer 1 Report

This sentence "Among the other substances contained in tea leaves, the beneficial effects are attributed mostly to Camellia sinensis, a plant rich in catechins – molecules with a positive effect on endothelial function [64]. The benefit of this product in the meta-analysis of Arab et al has been shown for both Asians and Non-Asians [26]."

Maybe, is better explained on this way:  Among the other substances contained in tea (Camellia sinensis) the beneficial effects are attributed mostly to catechins. Catechins are molecules with a positive effect on endothelial function [64]. The benefit of this product in the meta-analysis of Arab et al has been shown for both Asians and Non-Asians [26].

Please, include comment on recent work:

https://jamanetwork.com/journals/jamainternalmedicine/article-abstract/2749350

Author Response

Dear Editor,

Enclosed please find the manuscript formerly entitled “ Overview of meta-analyses: the impact of dietary lifestyle on stroke risk” by Altobelli E, Paolo Matteo Angeletti, Leonardo Rapacchietta, Reimondo Petrocelli.

We have highlight in green all changes requested by reviewer 1.

We hope that the revised manuscript may be acceptable for publication in International Journal of Environmental Research and Public Health.

Sincerely   

Emma Altobelli
